# Preparation of Molybdenum Disulfide with Different Nanostructures and Its Adsorption Performance for Copper (Ⅱ) Ion in Water

**DOI:** 10.3390/nano13071194

**Published:** 2023-03-27

**Authors:** You-Zhi Yao, Yong-Jie Shi, Kun-Hong Hu

**Affiliations:** 1School of Materials Engineering, Wuhu Institute of Technology, 201 Wenjin Rd., Wuhu 241003, China; yaoyz@whit.edu.cn; 2School of Energy Materials and Chemical Engineering, Hefei University, 99 Jinxiu Avenue, Hefei Economic and Technological Development Zone, Hefei 230601, China; 1937291899@qq.com

**Keywords:** copper (Ⅱ) ion, wastewater, molybdenum disulfide, adsorbent

## Abstract

The environmental problems in the world are attracting increasing amounts of attention, and heavy metal pollution in the water has become one of the focuses of the ecological environment. Molybdenum disulfide (MoS_2_) has excellent adsorption performance because of its extremely high specific surface area and unique active site structure, which has attracted an increasing amount of attention in the field of heavy metal disposal in various types of water. In this paper, two sorts of MoS_2_ nanoparticles, spherical and lamellar, were synthesized by different chemical methods. Their morphology and structure were characterized by scanning electron microscopy (SEM), transmission electron microscopy (TEM), X-ray diffraction (XRD), and a Raman spectrometer. The adsorption properties of two sorts of MoS_2_ nanoparticles for copper (Ⅱ) ions in water were investigated by changing the pH value, adsorption time, initial concentration of solution, adsorption temperature, etc. Finally, the adsorption mechanism was analyzed by kinetic, isothermal, and thermodynamic models. The results show that two microstructures of MoS_2_ nanoparticles can be used as efficient adsorption materials for removing heavy metal ions from water, although there are differences in adsorption capacity between them, which expands the theoretical basis of heavy metal adsorption in a water environment.

## 1. Introduction

Currently, two-dimensional nanomaterials have attracted great attention due to their excellent properties and wide range of applications. Among two-dimensional materials, graphene and transition metal two-dimensional materials, such as molybdenum disulfide (MoS_2_), have been studied the most [1]. MoS_2_ is very popular in the field of material research because of its wide source, low price, excellent photoelectric and mechanical properties, and unique layered structure [2]. MoS_2_ belongs to a class of transition metal disulfide compounds. There are three main crystal types, namely 2H, 3R and 1T-MoS_2_, where H, R and T represent hexagonal, rhombic, and tetragonal shapes, respectively, and the most common polycrystalline type is 2H-MoS_2_ [3]. The common structural forms of MoS_2_ nanomaterials mainly involve nanosheets, nanofilms, nanospheres, etc. [4,5]. The research results show that MoS_2_ has many excellent properties, such as no decomposition at 1200 degrees Celsius (°C), a specific surface area of up to 2630 m^2^/g, and a wide range of carrier mobility (μ ~700 cm^2^ V); furthermore, other atoms are easily implanted between its layers [6,7]. Layered MoS_2_ has inherent properties such as an adjustable band gap, high anisotropy, large interlayer distance, ultra-thin thickness, and interchangeable crystal form, which has currently drawn currenmore attention than graphene. The methods of MoS_2_ synthesis are numerous, including the liquid phase stripping method [8], magnetron sputtering method [9], hydrothermal method [10], micromechanical stripping method [11], mechanical grinding method, etc. [12]. MoS_2_ is generally used in electrocatalysis and lubrication. Fan [13] et al. prepared MoS_2_ with a flower shape by a hydrothermal method and found that its photocatalytic degradation of rhodamine B solution under visible light is very prominent. Xie [14] et al. prepared nano-MoS_2_ electrodes by the liquid-phase ultrasonic method and optimized the formation of MoS_2_-CuO_2_ composite electrodes, which significantly improved their electrochemical activity. Kun-hong Hu [15] studied the preparation of MoS_2_ nanoparticles/spent bleaching clay composite, which was used as a lubricating filler in ABS plastics to diminish friction.

Generally, adsorbents with excellent performance have the characteristics of large specific surface areas and rich functional groups [16]. Recently, MoS_2_ with the above properties has been widely explored as a new sorbent material. The growth of MoS_2_ on carbon nanofibers increased the specific surface area and the adsorption capacity by five times, which greatly improved the role of the composite in water purification and expanded the application range of MoS_2_ [17]. Rich active sites of sulfur atoms can combine with a variety of heavy metal ions. The experiment of Fi-fi Ja’s team showed that MoS_2_ had more active sites to absorb mercury (Ⅱ) ions in the water, and the surface activity was improved after heat treatment, which enhances the affinity of MoS_2_ to mercury. The experimental results showed that the adsorption of MoS_2_ on heavy metal ions has fast kinetic characteristics and excellent selectivity [18]. The MoS_2_/MXene nanocomposites synthesized by Asif Shahzad [19] also confirmed the excellent adsorption characteristics of MoS_2_ for heavy metal ions. Song-rong Li et al. [20] prepared a nanosheet N-doped carbon intercalated MoS_2_ nano-hybrid (NC-MoS_2_) with good interconnection by the one-pot hydrothermal method. The prepared composite was used as a new adsorbent for removing tetracycline (TC) in aqueous solution. Wei Wang [21] synthesized a new type of MoS_2_ montmorillonite hydrogel crosslinked by flower-shaped products and organic reagents (MoS_2_@2DMMT) under a hydrothermal reaction at 220 °C for 6 h. It was used to remove copper (Ⅱ) ions (Cu^2+^) in wastewater and inhibit bacterial growth.

The content of copper ions in the water affects human life and health. The concentration of Cu^2+^ in chromium or acid/alkali wastewater discharged from industrial production could be as high as 35 mg/L [22], while the standard value of the maximum allowable emission level of Cu^2+^ in the environment is only 0.25 mg/L [23]. If the concentration of Cu^2+^ in daily drinking or surface water is too high, or the content of Cu^2+^ in food is high after absorption and enrichment, which can cause various diseases, such as anemia, osteoporosis, and even liver cancer [24]. There are many methods to remove heavy metal Cu^2+^ from water, including membrane separation [25], electrochemical methods [26], chemical precipitation [27], adsorbent adsorption [28,29,30,31], etc. Among them, the sorption method, with the advantages of high efficiency, fewer by-products and simple equipment, has been widely employed in water treatment [32]. Pursuing adsorption materials with higher removal efficiency and less secondary pollution is still an important topic worth studying [33].

Hitherto, there have been few papers on Cu^2+^ sorption by different morphologies of MoS_2_. In this paper, two types of MoS_2_ nanoparticles were prepared by chemical and physical routes, and Cu^2+^ solutions with different concentrations were prepared with copper sulfate pentahydrate (CuSO_4_·5H_2_O). The nanoparticles were put into the prepared solution to be examined for static adsorption. The influence of changing adsorption factors was explored, and the influence of morphology on the adsorption amount was discussed. The adsorption process and mechanism were analyzed using a pseudo-second-order kinetic model. In the adsorption isotherm model, the Freundlich model is used to describe the relationship between its adsorption capacity and the concentration of the adsorbed solution. The value of the parameter n shows the adsorption performance of MoS_2_ for Cu^2+^. The results indicate that two types of MoS_2_ nanoparticles are excellent adsorption materials for heavy metal ions. Under the same initial concentration, pH value, temperature, and amount of adsorbent added, the adsorption performance of spherical MoS_2_ for Cu^2+^ is superior to that of lamellar MoS_2_. This result has not been reported in other literature, providing a new direction for exploring the synthesis and application of new spherical MoS_2_ nanocomposites. The brief processes of the experiment and adsorption are shown in Figure 1:

## 2. Experiment

### 2.1. Reagents and Materials

Sodium molybdate (Na_2_MO_4_·2H_2_O), thioacetamide (CH_3_CSNH_2_), and sodium sulfide (Na_2_S·9H_2_O) were procured from Shanghai Aladdin Biochemical Technology Co., Ltd. (Shanghai, China) Hydrochloric acid (HCl, 12 mol/L) was procured from Shanghai Bohe Fine Chemicals Co., Ltd. Anhydrous ethanol (C_2_H_5_OH) was procured from Tianjin Damao Chemical Reagent Factory (Tianjin, China) CuSO_4_·5H_2_O was procured from Sinopharm Chemical Reagent Co., Ltd. (Shanghai, China) HNO_3_ was procured from Wuxi Prospect Chemical Reagent Co., Ltd. (Wuxi, China). Sodium bicarbonate (NaHCO_3_) was procured from Tianjin Yongda Chemical Reagent Co., Ltd. (Tianjin, China). All of the above reagents were analytically pure and were used without further purification. H_2_SO_4_ (95–98%) was procured from China Suzhou Chemical Reagent Co., Ltd. (Suzhou, China). High-purity nitrogen (N_2_), ≥99.999%, was procured from Nanjing Shangyuan Industrial Gas Plant (Nanjing, China). The experimental deionized water (18.2 Ω) was manufactured by Milli-Q Direct 16 (Shanghai, China).

### 2.2. Instruments and Equipment

The X-ray diffractometer (XRD), a model D/max-RB, was manufactured by the Rigaku company (Akishima, Japan). The scanning electron microscope (SEM), a model SU8010, was manufactured by the Hitachi company (Ibaraki, Japan). The transmission electron microscope (TEM) was a Hitachi model H-800. The Raman spectrometer, Invia, was manufactured by the Renishaw company (Wotton-under-Edge, UK). The vacuum tube furnace, a model OTF-1200X-S, was manufactured b Hefei Kejing Material Technology Co., Ltd. (Hefei, China). The electrothermal constant temperature blast drying oven, a model DHG-9076A, was manufactured by Shanghai Jinghong Experimental Equipment Co., Ltd. (Shanghai, China). The CNC ultrasonic cleaner, a model KQ2200DB, was manufactured by Kunshan Ultrasonic Instrument Co., Ltd. (Kunshan, China). The atomic absorption spectrophotometer, a model AA900 (America) and the high-speed centrifuge, a model HC-2064, were manufactured by the Anhui Zhongke Zhongjia Scientific Instrument Co., Ltd. (Hefei, China).

### 2.3. Preparation of MoS_2_ with Different Micromorphologies

The two micromorphologies of nanomaterials were synthesized by optimizing the method reported by our research team earlier [34,35].

Nanospherical MoS_2_ was synthesized according to the following steps. A total of 3.00 g thioacetamide (TAA) and 1.50 g of sodium molybdate crystal (Na_2_MoO_4_·2H_2_O) were put into a 250 mL three-necked flask with 100 mL deionized water. The mixture was heated in a water bath while stirred. When the temperature reached 82 °C (not lower than 80 °C), 22.5 mL of anhydrous ethanol was added to the mixture. Subsequently, when the temperature was around 82 °C again, 33.8 mL HCl (12 mol/L) was put in the mixed solution; the reaction lasted for about 10 min (81 °C) and was terminated. After cooling, the yellowish-brown precipitate was removed from the flask, filtered, centrifuged, and washed with anhydrous alcohol and ultrapure water three times, respectively, to remove residual organic and inorganic substances. The experiments above were conducted in the fume hood throughout the process. The precipitate was put into a drying oven, then calcined in a vacuum tube furnace (100 °C, 2 h). The dried precipitate was calcined at 420 °C for 30 min under a flow of high-purity (99.999%) N_2_. After the desulfurization was complete, the pattern was ground with a ball mill for about 15 min to obtain the target product ①, which was labeled as S-MoS_2_.

Nanolamellar MoS_2_ was prepared according to the steps below. A total of 3.4 g sodium sulfide crystal (Na_2_S·9H_2_O), 0.6 g Na_2_MoO_4_·2H_2_O, and deionized water (100.0 mL) were placed in a three-necked flask. After stirring for 40 min at room temperature, 10.0 mL anhydrous ethanol was added into the three-necked flask. Stirring was continued, and 4.0 mL HCl (12 mol/L) was quickly added to the solution after about 10–20 min. At the instant moment, precipitation was observed. The following experimental process was similar to the preparation of S-MoS_2_, and the final target product obtained ② was labeled as L-MoS_2_.

### 2.4. Measurements and Analysis Approach of Cu^2+^ Removal Performance

An appropriate amount of CuSO_4_·H_2_O was placed in deionized water to fully dissolve, which was prepared with various concentrations of Cu^2+^ for standby. At a certain temperature and pH value, the absorbance was measured by an atomic absorption spectrometer after the solution stewed for 8 h, and the adsorption capacity of MoS_2_ nanoparticles on Cu^2+^ was calculated by the following formula:(1)$$q=\frac{C_{0}-C_{e}}{M}\times V$$

C_0_ represents the initial concentration of Cu^2+^ (mg/L), Ce represents the concentration of Cu^2+^ after adsorption equilibrium (mg/L), M represents the mass of MoS_2_ nanoparticles (g), and V represents the volume of adsorption solution (L).

Adsorption kinetic models have the significance of studying the relationship between concentration, adsorption time, and equilibrium adsorption capacity. In order to understand the control steps of the adsorption rate of MoS_2_ with different nanostructures, the pseudo-first-order equation [17]:(2)$$\ln\left(q_{e}-q_{t}\right)={\mathrm{lnq}}_{e}-K_{1}\;t$$
and pseudo-second-order equation [17]:(3)$$\frac{t}{q_{t}}=\frac{1}{K_{2}\cdot q_{e}^{2}}+\frac{1}{q_{e}}\cdot t$$
were used to analyze these relevant experimental data on the adsorption capacity to Cu^2+^ at different times [36]. Here, q_e_ (mg/g) represents the adsorption capacity of MoS_2_ on Cu^2+^ at adsorption equilibrium, q_t_ (mg/g) represents the adsorption capacity of MoS_2_ on Cu^2+^ at some time, K_1_ (h^−1^) represents the pseudo-first-order rate constant, and K_2_ (g/(mg∙h)) represents the pseudo-second-order rate constant.

Adsorption isotherms are very important for understanding the adsorption system between the adsorbate and adsorbent [37]. In order to further explore the interaction between Cu^2+^, S-MoS_2_, and L-MoS_2_ in the adsorption process, Langmuir and Freundlich isotherm models were generated to analyze the adsorption data. The formulas are as follows:

Langmuir adsorption isotherm model formula:(4)$$\frac{C_{e}}{q_{e}}=\frac{C_{e}}{Q_{m}}+\frac{1}{Q_{m}K}$$

Freundlich adsorption isotherm model formula:(5)$$\log q_{e}=\log K_{F}+\frac{1}{n}\log C_{e}$$
where C_e_ is the equilibrium concentration of Cu^2+^, mg/L; Q_e_ and Q_m_ are the adsorption capacity and saturated adsorption capacity, mg/g; and K (L/mg) is the Langmuir constant. K_F_ (L/mg) is a constant in the Freundlich adsorption isotherm model, which can reflect the saturated adsorption amount; 1/n is an empirical parameter expressing the adsorption strength. The adsorption strength in the Freundlich adsorption isotherm model changes with the material’s heterogeneity [38]. The adsorption isotherm model has a wide range of research on adsorption mechanisms, which is applicable to both single-layer adsorption and multi-layer adsorption. Therefore, the adsorption isotherm model is usually used to analyze the adsorption mechanism in the sorption process.

The temperature environment of adsorption will have an important impact on the adsorption effect. In order to deeply study how the temperature affects the effect of S-MoS_2_ and L-MoS_2_ on the Cu^2+^ in wastewater, thermodynamic parameters are used to express it. The changes in standard Gibbs free energy (ΔG, kJ/mol), standard enthalpy (ΔH, kJ/mol) and standard entropy (ΔS, J·mol·K^−1^) are calculated to determine whether adsorption is spontaneous, endothermic, or exothermic.

The adsorption thermodynamics formula [37] is as follows: (6)ΔG=−RTln(Kc)
(7)$$\ln(\mathrm{Kc})=\frac{\Delta S}{R}-\frac{\Delta H}{\mathrm{RT}}$$
(8)$$\mathrm{Kc}=\frac{\mathrm{Qe}}{\mathrm{Ce}}$$

R represents the molar gas constant (8.314 J·K·mol^−1^); T represents temperature (K); and Kc represents the thermodynamic constant. 

## 3. Results and Discussion

### 3.1. Characterization

The surface morphology of the two prepared products was characterized by SEM, as shown in Figure 2. In Figure 2a, it can be seen that the target product ① is a spherical particle. In Figure 2d, it can be seen that the target product ② shows a lamellar structure. In order to observe and analyze the morphology and structure more clearly, the two target products were further characterized by TEM. As shown in Figure 2b, s-MoS_2_ is a hollow spherical structure with an average particle size of about 50 nm. The spherical surface has numerous disorderly layered structures that can be observed after further amplification (Figure 2c). Figure 2e shows that the L-MoS_2_ structure is alternately stacked; the thickness of 8-layer MoS_2_ is about 5.1 nm after local magnification, and the layer spacing is about 0.635 nm, which is larger than that of commercial MoS_2_ (about 0.615 nm) [39], as shown in Figure 2f. Moreover, there are defects in the lamellar structure. Larger interlayer spacing and interlayer defects can increase the active sites of the adsorbent and reduce the ion diffusion distance [40]. The analysis results of SEM and TEM show that spherical and lamellar MoS_2_ were successfully synthesized.

The composition and crystal orientation structure of S-MoS_2_ and L-MoS_2_ were analyzed by XRD, as shown in Figure 3a. The peak positions of S-MoS_2_ and L-MoS_2_ were basically the same; the characteristic diffraction peaks appear at 13.9°, 33.58°, 39.55° and 58.66° at 2 θ, which correspond to the four crystal planes (002), (100), (103) and (110), respectively. The crystal planes are consistent with the MoS_2_ standard card JCPDS37-1492 with a hexagonal crystal structure [41]. It can be inferred that the substances prepared by the two different methods are both MoS_2_. No other peaks appear in the XRD spectrum, indicating that the impurities contained in the nanocompound are very small.

The Raman spectra were further used to verify the composition of the two nanocompounds, as shown in Figure 3b. It can be clearly seen that there are two main Raman active peaks in the two spectral lines, namely 379.3 cm^−1^ and 405 cm^−1^, corresponding to the E^1^_2g_ mode and the A_1g_ mode of MoS_2_ [42]. The results show that the components of the synthesized material obtained by the two methods are both MoS_2_.

### 3.2. Factors Affecting Adsorption Performance

#### 3.2.1. Effect of Initial pH Values

The pH values of the solution could influence the existing speciation of Cu^2+^ and its activity point. The Ksp of copper hydroxide at 25 °C is 2.6 × 10^−19^, and 60 mg/L Cu^2+^ solution can generate precipitation when the pH value is higher than 6 [30]. Hence, the adsorption performance of Cu^2+^ onto MoS_2_ at different pH values was assessed. As shown in Figure 4, the adsorption capacity of MoS_2_ nanoparticles for Cu^2+^ increases continuously at pH values from 2 to 5.

At low pH values, the adsorption capacity is small, which may be attributed to the large degree of protonation of functional groups on the surface of MoS_2_ nanoparticles. H^+^ occupies a large number of adsorption sites that should belong to Cu^2+^, reducing the number of sites that coordinate with Cu^2+^, resulting in the reduction of the adsorption capacity of Cu^2+^ [43]. With the increase in pH values, the degree of protonation decreases, more active sites of S atoms are released, and the binding rate of Cu^2+^ increases. As the pH value exceeds 5, the adsorption capacity decreases, which may be due to the combination of some Cu^2+^ and OH^−^. Therefore, the two types of MoS_2_ achieve the best adsorption performance at pH 5.

#### 3.2.2. Effect of Adsorption Duration

The optimum equilibrium time of adsorption was investigated at 25 °C, C_0_ 60 mg/L. With the increase in the adsorption duration, the adsorption capacity for Cu^2+^ gradually increased, and the adsorption reached equilibrium at 8 h (shown in Figure 5). The adsorption capacity of L-MoS_2_ increased faster from 0 h to 2 h. However, the adsorption capacity of S-MoS_2_ nanoparticles changed significantly and was larger. Generally, in the initial adsorption stage, there are many active sites on the surface and pores of MoS_2_. Under the action of diffusion and electrostatic attraction, Cu^2+^ combines with the active sites rapidly, thus showing the characteristics of a fast adsorption rate [44]. With the lapse of time, the decrease in the adsorption capacity may be attributed to the reduction of adsorption sites, the weakening of diffusion dynamics, and surface electrostatic repulsion [45]. After the two kinds of adsorption reach equilibrium, the reason for the slight decrease may be that a small amount of desorption occurs. Consequently, the best adsorption time of Cu^2+^ on MoS_2_ with its two morphologies is 8 h.

#### 3.2.3. Effect of Initial Concentration

The initial concentration of ions in the wastewater has a significant impact on the adsorption effect [46]. Five different concentrations of Cu^2+^ solutions of 20 mg/L, 40 mg/L, 60 mg/L, 80 mg/L, and 100 mg/L were prepared in the experiment, and 40 mL of each solution was taken. Under the conditions of pH 5, 0.020 g spherical MoS_2_, 0.020 g lamellar MoS_2_, and stewing for 8 h, the change in the adsorption capacity of the two nanomaterials was assessed, and the result is shown in Figure 6. With the increase in initial concentration, the adsorption capacity for heavy metal Cu^2+^ in wastewater also gradually rises, but the removal efficiency gradually declines. q_S-MoS2_ > q_L-MoS2_ may be due to the difference in the microstructure, porosity, and adsorption dynamics of active interlayer sites, indicating that the material’s morphology has a significant impact on its adsorption capacity. The adsorption capacity reaches its maximum at 80 mg/L. According to the calculation of Formula (1), the adsorption capacity of S-MoS_2_ is 117.89 mg/g, and the adsorption capacity of L-MoS_2_ is 107.62 mg/g. Removal rates are 73.68%, 67.26%, respectively, indicating that the materials still have good adsorption performance. However, while at 100 mg/L, the adsorption capacity diminishes because of the large electrostatic repulsion between the ions adsorbed at the effective site [47].

#### 3.2.4. Effect of Different Temperatures

Figure 7 shows the effect of temperatures on the adsorption of Cu^2+^. The results showed that the adsorption capacity increased with an increase in temperatures (20–80 °C). This may be attributed to the accelerated diffusion rate and increased effective active sites [48]. When the adsorption temperature rises from 80 °C to 100 °C, the adsorption capacity decreases, which may be due to the destruction of adsorption equilibrium, the enhancement of the desorption process, and the intensification of molecular movement above 80 °C. The experimental results also show that the adsorption of Cu^2+^ by S-MoS_2_ and L-MoS_2_ can be carried out in a wide temperature range.

### 3.3. Adsorption Kinetics and Isotherm

#### 3.3.1. Adsorption Kinetics

The pseudo-first-order and pseudo-second-order adsorption kinetic models were performed to fit the adsorption process at a constant concentration (60 mg/L) and constant temperature (25 °C) within 0–24 h. Linear fitting of ln (q_e_–q_t_) and time was performed on the experimental data. According to Formula (2), the fitting results of the pseudo-first-order kinetic model for the adsorption of Cu^2+^ by S-MoS_2_ and L-MoS_2_ showed that the two linear regression coefficients R^2^ were 0.3696 and 0.4103, respectively. As we know, the linear correlation coefficient (R^2^) is greater than 8 to reflect a high degree of consistency between the model and the actual situation. Obviously, the actual adsorption of Cu^2+^ by MoS_2_ nanoparticles does not conform to the pseudo-first-order kinetic model.

According to Formula (3), the linear fitting curve of t and t/q using the pseudo-second-order models for the adsorption of Cu^2+^ on S-MoS_2_ and L-MoS_2_ are shown in Figure 8. The linear correlation coefficients R^2^ after the fitting of the adsorption process are 0.981 and 0.965, which is much greater than 0.8, indicating that adsorption processes on Cu^2+^ in wastewater corresponded to the pseudo-second-order kinetic model. It can be inferred that the entire adsorption process is dominated by chemical adsorption, supplemented by physical adsorption [49].

#### 3.3.2. Adsorption Isotherm

At 25 °C, according to the adsorption equilibrium concentration and saturated adsorption capacity of Cu^2+^ solution with different initial concentrations, the Langmuir adsorption Formula (4) and Freundlich Formula (5) were used to analyze the data, as shown in Figure 9. Evidently, the fitted plots of the Langmuir model are closer to the experimental curves than those of the Freundlich model.

The corresponding slope and intercept were calculated by linear fitting in Figure 9, and the important parameters in the Langmuir adsorption isotherm model (Figure 9a) and Freundlich adsorption isotherm model (Figure 9b) were obtained, as shown in Table 1. The constants (K) of S-MoS_2_ and L-MoS_2_ displayed by Langmuir are 0.0913 L/mg and 0.0442 L/mg, and the correlation coefficients (R^2^) are 0.9842 and 0.9695, respectively. The constants (K_F_) displayed by the Freundlich model are 14.39 and 12.11 (L/mg), and the correlation coefficients R^2^ are 0.9100 and 0.9307, respectively. The higher value of Langmuir’s correlation coefficient R^2^ indicates that the Langmuir model fits the adsorption isotherm better than the Freundlich model, which proves the monolayer adsorption mechanism. The calculated values of Cu^2+^ adsorbed by the two nanomaterials are similar, reaching about 136 mg/g.

Nevertheless, if the adsorption isotherms of two nanomaterials are analyzed at low concentrations (20–80 mg/L), the higher value of the Freundlich correlation coefficient R^2^ indicates that the change in the adsorbent surface is involved in the multi-layer diffusion on the heterogeneous adsorption site, and the adsorption of Cu^2+^ by S-MoS_2_ and L-MoS_2_ is a multi-molecular layer adsorption process. The value of n in this experiment is greater than 0.5, which indicates that S-MoS_2_ and L-MoS_2_ have good adsorption performance for Cu^2+^ [50].

Table 2 lists the comparison of Cu^2+^ adsorption capacity between S-MoS_2_, L-MoS_2_, and the materials reported previously. In this work, the adsorption capacity of synthetic materials for Cu^2+^ is superior to most other adsorbents. This can be attributed to the layered structure and surface defects of the material, which contribute to the capture of Cu^2+^. Compared to MoS_2_ nanosheets, our synthetic materials have a low adsorption capacity, which may be due to the thin MoS_2_ layer (1–2 layers) synthesized by Huang et al., which is more conducive to the adsorption of Cu^2+^.

#### 3.3.3. Thermodynamic Analysis

From 293 K to 373 K, a Van’s t-Hoff diagram of Cu^2+^ adsorption is formed according to Formulas (6)–(8), as shown in Figure 10.

The values of ΔH and ΔS are calculated by the slope and intercept of the fitting diagram, as shown in Table 3 below.

The correlation coefficients R^2^ of the thermodynamic model are 0.883 and 0.881, respectively, which are greater than 0.8, indicating that they have a good correlation. ΔG < 0 indicates that the process of adsorption is spontaneous. ΔG decreases with increasing temperature, which indicates that the higher the adsorption environment temperature is, the better the adsorption effect will be. 

When ΔH falls into a range of 2.1–20.9 kJ/mol, adsorption involves mainly a physical process [53]. The ΔH values of S-MoS_2_ and L-MoS_2_ are both greater than 20 kJ·mol^−1^, between 20.9–418.4 kJ·mol^−1^, indicating that the process is dominated by chemical adsorption [54]. ΔH > 0 reveals that adsorption is an endothermic behavior [55]. The heat absorption in the process may involve electrostatic adhesion and the formation of chemical bonds between Cu^2+^ and sulfur atoms (S) on the surface of S-MoS_2_ and L-MoS_2_.

## 4. Conclusions

In summary, using sodium molybdate crystals as precursors, we prepared MoS_2_ with spherical and layered structures by reacting with different reactants, calcining, and milling. The adsorption properties of two types of MoS_2_ were carried out by removing Cu^2+^ from wastewater. The influence mechanism and microstructure of S-MoS_2_ and L-MoS_2_ adsorption were measured by changing environmental factors such as pH, temperature, concentration, and adsorption time. The results show that the adsorption capacity of S-MoS_2_ is higher than that of L-MoS_2_ under the conditions of pH 5, 80 °C, 80 mg/L, and 8 h, indicating that the morphology has an impact on the adsorption performance of the material. The theoretical adsorption capacity of the two nanomaterials might reach about 136 mg/g. The fitting of Langmuir and Freundlich adsorption isotherm models for different concentration ranges is high, and it can be inferred that the adsorption of Cu^2+^ on S-MoS_2_ and L-MoS_2_ particles has a mixed control process of surface adsorption and internal fine pore adsorption. The thermodynamic results show that the adsorption of Cu^2+^ by different micromorphologies of MoS_2_ nanoparticles is a spontaneous endothermic process. These results might provide a theoretical reference for the further development of various structures of MoS_2_ as adsorbents with excellent performance.

## Figures and Tables

**Figure 1 nanomaterials-13-01194-f001:**
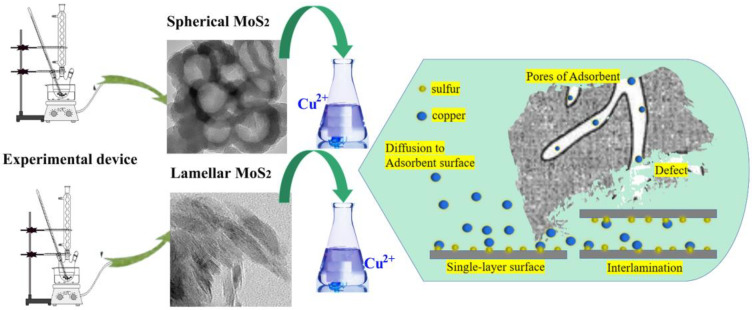
Schematic of experimental and adsorption processes.

**Figure 2 nanomaterials-13-01194-f002:**
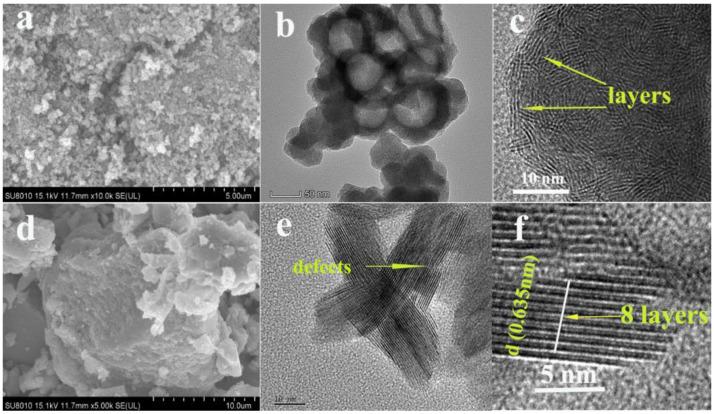
SEM (**a**,**d**) and TEM (**b**,**c**,**e**,**f**) images of the spherical MoS_2_ and lamellar MoS_2_.

**Figure 3 nanomaterials-13-01194-f003:**
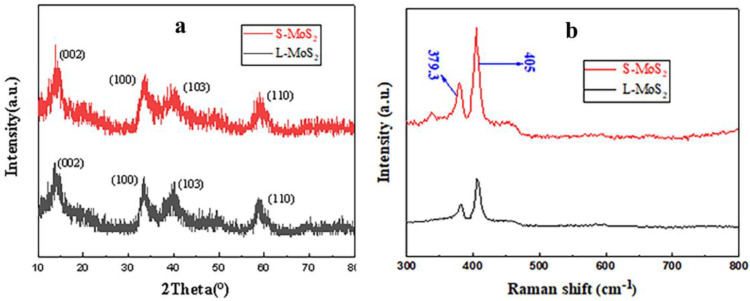
XRD (**a**) and Raman (**b**) spectra of the S-MoS_2_ and L-MoS_2_.

**Figure 4 nanomaterials-13-01194-f004:**
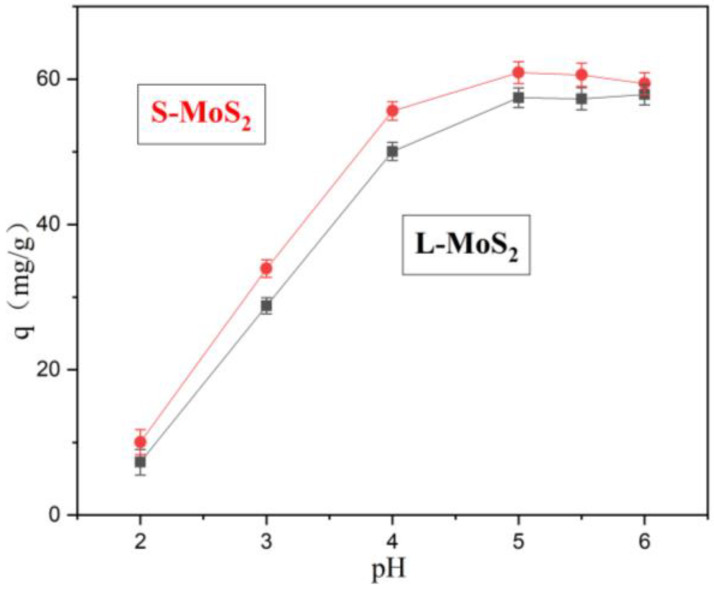
Cu^2+^ adsorption onto MoS_2_ with two morphologies at different pH values (C_0_: 60 mg/L, Time: 8 h, 25 °C).

**Figure 5 nanomaterials-13-01194-f005:**
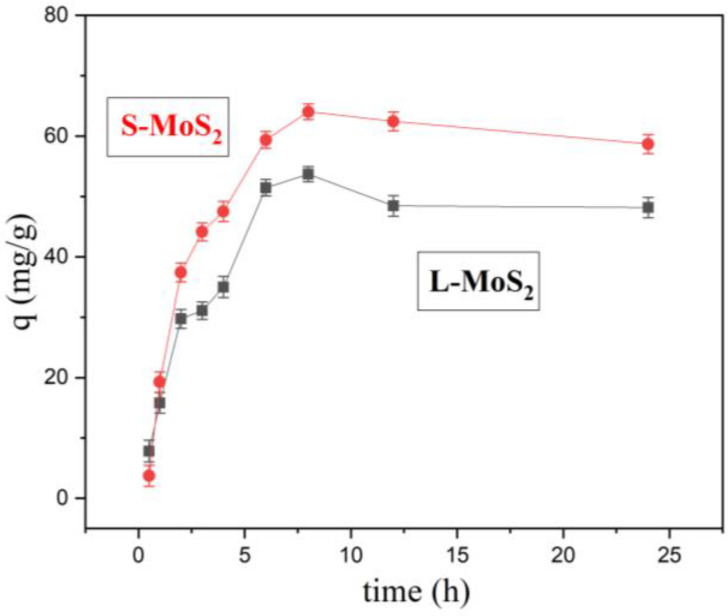
Cu^2+^ adsorption capacity at different times (C_0_: 60 mg/L, pH 5).

**Figure 6 nanomaterials-13-01194-f006:**
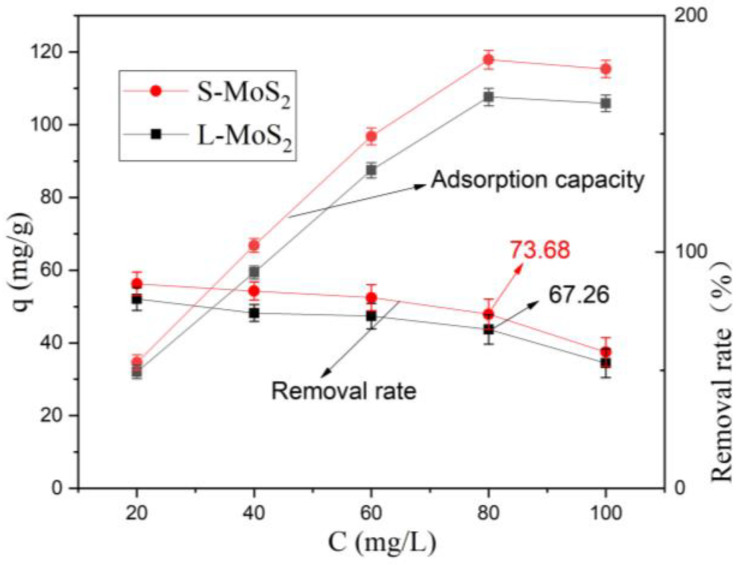
Effect of different initial concentrations (Time: 8 h, pH 5, 25 °C).

**Figure 7 nanomaterials-13-01194-f007:**
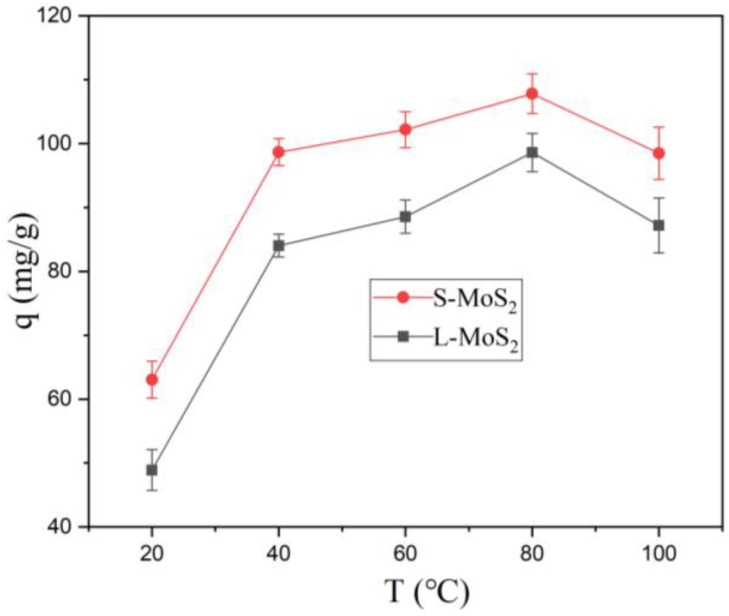
Effect of different temperatures (C_0_: 60 mg/L, time: 8 h, pH 5, 25 °C).

**Figure 8 nanomaterials-13-01194-f008:**
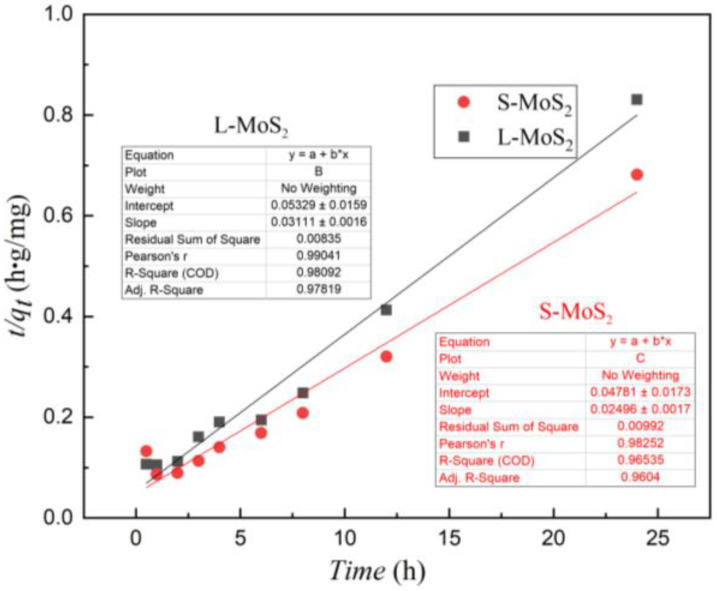
Pseudo-second-order kinetic models for S-MoS_2_ and L-MoS_2_ adsorption of Cu^2+^ (The inserted chart: the parameter of the linear fitting).

**Figure 9 nanomaterials-13-01194-f009:**
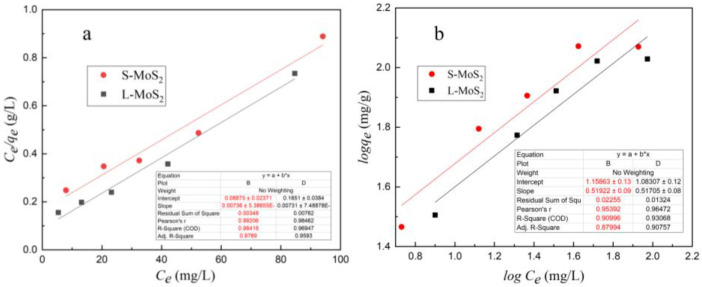
Langmuir (**a**) and Freundlich (**b**) isotherm models fitting Cu^2+^ adsorption onto S-MoS_2_ and L-MoS_2_ (20–100 mg/L) (The inserted chart: the parameter of the linear fitting).

**Figure 10 nanomaterials-13-01194-f010:**
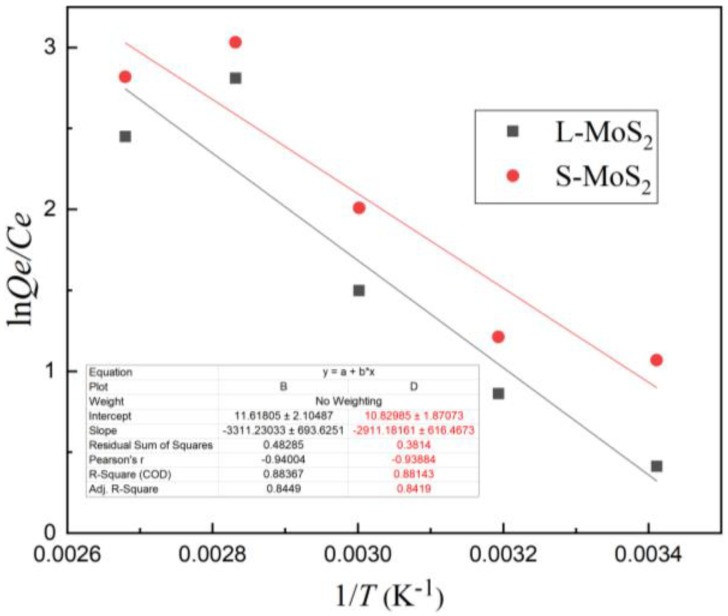
Thermodynamic model fitting for adsorption of Cu^2+^ on S-MoS_2_ and L-MoS_2_ at different temperatures.

**Table 1 nanomaterials-13-01194-t001:** Comparison of the maximum copper (II) ions adsorption capacities for various MoS_2_ adsorbents from Langmuir isotherm.

Langmuir Adsorption Isotherm Model				Freundlich Adsorption Isotherm Model		
MoS_2_	K (L/mg)	Q_m_ (mg/g)	R^2^	n	K_F_	R^2^
S-MoS_2_	0.0913	136.98	0.9842	1.99	14.39	0.9100
L-MoS_2_	0.0442	136.79	0.9695	1.93	12.11	0.9307

**Table 2 nanomaterials-13-01194-t002:** Comparison of the maximum copper (II) ions adsorption capacities for various MoS_2_ adsorbents from Langmuir isotherm.

Adsorbent	Q_m_ (mg/g)	Experimental Conditions		Ref
		pH	Temperature (°C)	
MoS_2_ nanosheets	169.5	6	25	[51]
MoS_2_@2DMMT	65.75	5	30	[21]
Polypyrrole-MoS_4_	111	3	room temperature	[52]
MoS_2_@PDA@PAM	50.57	7	25	[38]
Spherical MoS_2_	136.98	5	25	This work
Lamellar MoS_2_	136.79	5	25	This work

**Table 3 nanomaterials-13-01194-t003:** Related parameters of adsorption thermodynamic model at different temperatures.

MoS_2_	T (°C)	ΔG (kJ/mol)	ΔS (J·K/mol)	ΔH (kJ/mol)	R^2^
L-MoS_2_	20	−1.01	96.592	27.529	0.881
-	40	−2.24	-	-	-
-	60	−4.148	-	-	-
-	80	−8.248	-	-	-
-	100	−7.601	-	-	-
S-MoS_2_	20	−2.605	90.041	24.203	0.884
-	40	−3.155	-	-	-
-	60	−5.561	-	-	-
-	80	−8.900	-	-	-
-	100	−8.742	-	-	-

## Data Availability

All data generated or analyzed during this study are included in this published article.

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
