# Peer review of "Preparation of Molybdenum Disulfide with Different Nanostructures and Its Adsorption Performance for Copper (Ⅱ) Ion in Water"

_nanomaterials, 2023, doi:10.3390/nano13071194_

Round 1

Reviewer 1 Report

In general, the article covers a fairly important topic, but the results are presented insufficiently, and the article needs to be revised before publication.

1.            The introduction should provide more background information to ensure that readers are not assumed to be familiar with the topic. The quasi-second-order kinetic model mentioned in the text should be described more thoroughly, either in the main text or in the supplementary materials, and links to relevant sources should be provided.

2.            The text should include references to previous studies, such as the MoS2 standard card JCPDS37-1492, to allow readers to consult them if necessary.

3.            The text should provide a more detailed discussion of the implications of the findings and potential future directions for research, including the potential use of obtained results for other synthesis conditions layered materials.

4.            The limitations and potential biases in the methodology used should be discussed in Part 3 or in the conclusions to provide a more critical evaluation of the results.

5.            The strong deviation from linear behavior in Figure 10 should be discussed in more detail, and the appropriateness of such an approximation in this case should be addressed.

6.            Molybdenum disulfide has several phases (2H, 3R, 1T) [https://www.sciencedirect.com/science/article/pii/S0927025618303628]]. It would be useful to know whether the phase of molybdenum disulfide observed in the synthesized samples was analyzed. This information would provide important insights into the nature of the synthesized samples

7. The text should be improved by providing more details on the repeatability of the synthesis of the samples and whether the results would still be valid in the case of slightly changed synthesis conditions.

Reviewer 2 Report

Thank you for sending this work entitled “Preparation of Molybdenum Disulfide with Different Nanostructures and Its Adsorption Performance for Copper (â…¡) Ion in Water”. I recommend this paper for publication after minor revision which should be corrected before publishing this manuscript.

 1-    There are some grammatical and spelling mistakes. These mistakes should be revised.

2-    For example: in the caption of Fig 6. It should be pH values

3-    Figure 5 is missed

4-    No caption for Fig 9

5-    It was better if the authors compare the performance of the prepared molybdenum disulfides with other reported adsorbents to show the efficiency.

Reviewer 3 Report

The manuscript proposed by the authors is of interest and deserves to be published with significant corrections. First of all, the authors demonstrated a good mastery of the production of different forms of MoS2 usable as adsorbent material. This part of the study is quite original. The study of the adsorption potential of copper in solution by MoS2 nanoparticles is quite conventional since there are hundreds of publications on the adsorption capacity of different synthetic and natural materials for the fixation of different metal ions.

In order to demonstrate the interest of these nanoparticles for the adsorption of copper, it would be essential to compare the results obtained with the results presented in other scientific publications. Adding a table for this would be advisable.

From the point of view of form, the manuscript could be greatly improved. Here are some examples of suggestions:

1) in order to reduce the number of figures, figures 3 and 4 could be combined into one (fig. 3a and 3b);

2) note that there is no figure 5;

3) several pieces of text presented in the Results and discussion section should be placed in the Materials and methods section (e.g. lines 165-172, 239-246, 262-276, 308-316);

4) add information regarding MoS2 concentration, temperature and reaction time in the title of Figure 6;

5) add information about MoS2 concentration and pH in the title of Figure 7;

6) add information about MoS2 concentration, temperature, time and pH in the title of Figure 8;

7) add a title to Figure 9 and enter information regarding the conditions of pH, time, MoS2 concentration and initial copper concentration;

8) figure 10 is not very useful and could be removed;

9) the authors could choose only one figure among figures 12 and 13;

10) were the adsorption tests carried out in triplicate? If yes, add error bars in the figures;

11) Replace the "s" and "l" in the legend of figure 14 by "S" and "L".

Round 2

Reviewer 1 Report

Please double-check your grammar and syntax (e.g. double dots on line 102 and similar typos). On the whole, the article has been improved and can be published after minor revision.

Author Response

Dear editors and referees:

According to your request, we revised our manuscript and thank you to allow us to resubmit the revised version. Here we simply described our revision.

All modifications have been highlighted in yellow. They are located on lines 11, 13-15, 20, 24, 34, 37, 46, 48, 59, 61, 65, 72, 77, 79, 80, 87, 100, 113, 123, 124, 192, 193, 195, 231-236, 238, 242, 279, 285-287, 292, 293, 298, 331, 332, 337, 348-354, 381-394 of the text.

References have also been adjusted and revised.

Thank the editors and referees very much.

Reviewer 3 Report

The authors have made the requested changes. I think the manuscript is acceptable in its current form.
